# Transformational Leadership, Achievement Motivation, and Perceived Stress in Basic Military Training: A Longitudinal Study of Swiss Armed Forces

**Sandra Sefidan** [1,2], **Maria Pramstaller** [1,2,3], **Roberto La Marca** [1,4], **Thomas Wyss** [5], **Lilian Roos** [5], **Dena Sadeghi-Bahmani** [6,7,8], **Hubert Annen** [2,*] **and Serge Brand** [7,8,9,10,11]

1  Department of Clinical Psychology and Psychotherapy, University of Zurich, 8050 Zurich, Switzerland; s.sefidan@hotmail.com (S.S.); maria@pramstaller.org (M.P.); Roberto.LaMarca@clinica-holistica.ch (R.L.M.)
2  Military Academy, Swiss Federal Institute of Technology ETH Zurich, 8903 Birmensdorf, Switzerland
3  Praxis Pramstaller, Seestrasse 107, 8707 Uetikon am See, Switzerland
4  Clinica Holistica Engiadina, Centre for Stress-Related Disorders, 7542 Susch, Switzerland
5  Swiss Federal Institute of Sport Magglingen SFISM, 2532 Magglingen, Switzerland; thomas.wyss@baspo.admin.ch (T.W.); roos.lilian@gmail.com (L.R.)
6  Department of Psychology, Stanford University, Stanford, CA 94305, USA; bahmanid@stanford.edu
7  Center for Affective, Stress and Sleep Disorders (ZASS), Psychiatric University Hospital Basel, 4002 Basel, Switzerland; Serge.Brand@upk.ch
8  Sleep Disorders Research Center, Kermanshah University of Medical Sciences, Kermanshah 67146, Iran
9  Substance Abuse Prevention Research Center, Health Institute, Kermanshah University of Medical Sciences, Kermanshah 67146, Iran
10  School of Medicine, Tehran University of Medical Sciences, Tehran 25529, Iran
11  Division of Sport Science and Psychosocial Health, Department of Sport, Exercise and Health, University of Basel, 4052 Basel, Switzerland
*  Correspondence: hubert.annen@milak.ethz.ch

**Abstract:** In Switzerland, military service is a civic obligation for all adult male citizens, and thus, leadership style can be particularly challenging. The present study investigated the impact of superiors' leadership styles on recruits' achievement motivation, organizational citizenship behavior (OCB), and perceived stress during their Basic Military Training (BMT). To this end, a total of 525 male recruits (mean age: 20.3 years) recruits were assessed both cross-sectionally and longitudinally. At the start of BMT (baseline), at week 7, and at week 11, participants completed a series of self-rating questionnaires covering demographic information, achievement motivation, organizational citizenship behavior (OCB), perceived stress, and their superiors' leadership styles (transformational, transactional und laissez-faire). Longitudinally, scores for achievement motivation and OCB showed no significant difference between baseline and the 11th week. In a group comparison, the group experiencing higher transformational leadership (from week 7 to week 11) had the highest scores for achievement motivation and OCB, and the lowest scores for perceived stress, all at week 11. Exploratively, achievement motivation and OCB at baseline were associated with transformational leadership and transactional leadership at week 7 and week 11. Perceived stress at baseline correlated only with transformational leadership but not with transactional leadership, both at week 7 and week 11. Transformational leadership style fostered achievement motivation and OCB in Swiss military recruits and protected them from stress, both cross-sectionally and longitudinally.

**Keywords:** transformational leadership; transactional leadership; achievement motivation; organizational citizenship behavior; perceived stress

## 1. Introduction

Leadership is essential in the military context, improving subordinates' performance and minimizing the loss of resources [1,2]. Here, we first examine what is known about leadership in general and leadership in the military context. Then, we shift to the particular

case of the Swiss Armed Forces and its leadership culture. Finally, we discuss the impact of leadership on different performance variables of subordinates in the military environment and in our study in particular.

### 1.1. Leadership General Concepts

Leadership is a set of behaviors intended to instruct, guide, supervise, and empower employees; it is a learnable and teachable skill [3]. Not surprisingly, the training and management of successful leaders has become essential in the economic and military environments. Effective and successful leadership behaviors on the part of managers are key characteristics of a high-performing organization [4]. Due to common definitions and concepts of leadership (see [5]), the underlying mechanism of leadership influence on outcome variables is still divergent [6]. Published reviews [7,8] cover different constructs and theories, which are subject of intensive research. Various leadership concepts and definitions can be found in current literature such as: Collaborative leadership [9] and Entrepreneurial leadership [10]. However, research on leadership concepts has in common that the positive or negative influence of leadership styles and its effect on subordinates is of interest [11]. From this diversity of leadership concepts, the present study focused on the concept of "full range" leadership, particularly transformational leadership, from Bass [12]. Transformational leadership [12] is one of the most extensively studied constructs in this context [13]. The theory advanced by Bass [12] structures leadership in terms of a "full range of leadership", identifying complementary leadership styles, namely transformational, transactional, and laissez-faire leadership [14]. According to Bass [12], transactional leadership focuses on rewarding effort and commitment instead of changing followers' behavior. The reward of followers' performance corresponds to the agreed and aligned objectives on which achievements are measured. The transformational leader, however, influences or transforms the behaviors, values, and motives of employees to turn short-term, self-serving objectives into long-term high-level values and ideals [12]. Transformational leadership encourages employees to perform beyond expectations, and thereby results in a more successful leadership effect [4,12,15,16].

The work of Bass [12] points to an augmentation effect of transformational leaders over and above that of transactional leaders [4]. This augmentation effect is defined as increased effectiveness, satisfaction, performance, motivation, and effort on the part of subordinates [12] and has been replicated in several cross-sectional studies covering both military (e.g., [17] and civil [16]) contexts. However, in the former context there has, to date, been no examination of the augmentation effect in a longitudinal study design. The present study fills this gap.

### 1.2. Leadership in the Military Context

Military culture arises out of particular norms and history, making armies exceptional [18]. While there are parallels with civilian organizations, everyday military life differs with respect to the physical and psychological demands it presents. Greater challenge, stress, complexity, and uncertainty are present in military training and operations [19,20].

Similar features can be found in many armies, including hierarchical structure, formal management structures, conservatism, an emphasis on the status quo, and high formalization [21].

Several studies indicate leadership competency to be the main factor in the performance and success of military units (e.g., [22–24]). Competent military leaders can play a determining role in fostering a unit's quality, efficiency, and effectiveness [25], increase soldiers' commitment to their role, lead to trust, clarify intentions, inspire confidence, build teams, set an example, and keep hope alive [26,27]. Furthermore, they motivate subordinates to fight for one another on the battlefield [28] and increase willingness to put their lives in danger [29]. Given this, for military organizations, *transformational leadership* as a supportive style appears to be the most effective form of leadership [28]. However, promoting such a style in, as mentioned above, a strongly hierarchical, formal, and conservative

culture is a particular challenge. Providing evidence for its benefits is therefore helpful. Both the leadership style of military personnel in general and transformational leadership in particular have become important topics shaping military doctrine [30]. In this respect, the U.S. Army has already begun to investigate the distinct leadership styles identified in Bass' theory, with the aim of identifying implications for successful leadership [31]. The "full range of leadership model" of Bass [12] has proved to be seminal in research into high-reliability organizations such as the military (e.g., [21,32–34]).

### 1.3. Swiss Armed Forces: A Special Case

The Swiss Army is a conscript army or, by definition, a "training army" (in German: "Ausbildungsarmee"; SWISSINT, 2020). Military service is mandatory for all Swiss males (Federal Authorities of the Swiss Confederation, 2013a). Importantly, Swiss citizens are released from employment or studies to undertake military service (Basic Military Training (BMT) and annual refresher courses) for the period required [35]. Given the mandatory nature of this service, there is potential for recruits to be not entirely motivated to perform at the highest level during BMT [35].

### 1.4. Leadership in the Swiss Armed Forces

Not least to improve its image among the civilian population, the Swiss Armed Forces made repeated efforts to promote a more humane leadership style. Representative of this is the book "Menschenorientierte Führung (human-oriented leadership)" [36], which was also the official teaching manual from 1991 to 2004. Without explicitly referring to the concept of transformational leadership, connections are apparent in this manual with references to such matters as communicating enthusiastically, encouraging initiative, knowing and promoting subordinates' potential, and acting in an exemplary manner. This book was replaced by the manuals currently used to prepare military cadre candidates for a civilian leadership certificate. These also discuss the advantages and disadvantages of the various leadership styles [37].

At the same time, studies have sought to capture the effects of transformational leadership empirically. Thus, among other things, the augmentation effect has been confirmed [17] as well as the influence on organizational citizenship behavior [38] and trust [39]. Finally, the new "Army Vision 2030" (www.vision-armee.ch; accessed on 20 October 2021) has triggered targeted efforts to make transformational leadership explicitly part of the culture of the Swiss Armed Forces. To achieve this, the elements of transformational leadership are integrated into the regular employee appraisal system so that leaders must regularly reflect on themselves with regard to these aspects [40]. Against this backdrop, it is more than appropriate to evaluate the value and impact of transformational leadership and thus demonstrate to those in leadership positions the benefits of engagement with this style of leadership.

### 1.5. Impact of Transformational Leadership on Subordinates' Performance Variables in a Military Context

Research in the military environment confirms the positive impact of transformational leadership style on subordinates in several respects. For example, when subordinates perceive leadership to be more transformational, this results in stronger identification and internalization with respect to leaders [41]. Research has also shown that transformational leadership increases subordinate's military hardiness [42] and individual creativity [21]. Transformational leadership style can foster interpersonal and organizational relationships, and it creates a friendly atmosphere that increases satisfaction, motivation, and defense commitment [42].

Although a few previous studies have assessed the impact of transformational leadership on various outcomes in the military context (see above), the longitudinal impact of transformational leadership on achievement motivation, organizational citizenship behavior (OCB), and perceived stress of subordinates in the Swiss Armed Forces [43] has not as yet been addressed. Organizational citizenship behavior (OCB) refers to the in-

vestment of a person in the social environment in such a way as to enhance performance. OCB is defined as an individual behavior not explicitly specified in the formal reward system [44]. OCB entails extra effort that serves to increase the social and psychological capital of an organization by helping others [45], "without the expectation of immediate reciprocity" ([46], p. 151). In military research, relevant daily motivators of personnel act to prevent increased stress among subordinates [43]. The existing research literature is mainly based on cross-sectional studies. Stadelmann [17] found an augmentation effect on subordinates' voluntary extra effort in the Swiss Armed Forces. Kane and Tremble Jr [47] identified an augmentation effect on the job motivation and affective commitment of subordinates of the U.S. Army. Bass and Stogdill [48] theorized that leaders' increasing demands may result in higher stress for their employees. Given this, leadership style is expected to affect subordinates' stress because of its influence on stressors and the coping resources of employees [49]. In addition, Rowold and Schlotz [49] proposed that the supportive style of a transformational leader helps followers reframe stress-related events and helps them avoid stress. While the impact of transformational leadership on perceived stress of subordinates has been studied in numerous civilian organizations, there has been no corresponding work in military organizations towards perceived stress of subordinates during Basic Military Training.

The present research fills the research gap concerning both motivational aspects such as achievement motivation and organizational citizenship behavior (OCB) and perceived stress with a study of subordinates in the military environment of the Swiss Armed Forces.

Given this background, the aims of the present study were four-fold. First, we aimed to investigate the correlation of the study variables (achievement motivation and OCB) between baseline and week 11. Second, we sought to calculate to what extent a perceived alteration in transformational leadership style between the beginning and the 11th week of BMT has an impact on achievement motivation, OCB, and perceived stress assessed at week 11. Our third aim was to investigate the augmentation effect of transformational leadership style over and above transactional leadership style longitudinally at week 11. Fourth, we aimed to investigate the correlation between achievement motivation, OCB, and perceived stress at baseline and leadership style at week 7 and week 11 in an exploratory way. No previous study has made this kind of longitudinal investigation with Swiss Armed Forces recruits.

The following three hypotheses and one exploratory research question were formulated. First, based on [35,50,51] we expected that achievement motivation and OCB would show constant scores at baseline and week 11 of BMT. Second, also on the basis of these earlier studies [46,49,52–57], we expected that perceptions of an increased transformational leadership style from week 7 to week 11 would lead to higher achievement motivation, higher organizational citizenship behavior (OCB) and lower perceived stress in subordinates at week 11. Third, on the basis of other studies [4,17,47], we anticipated that a transformational leadership style would have an augmentation effect over transactional leadership style with respect to impact on achievement motivation, organizational citizenship behavior (OCB), and perceived stress in subordinates at week 11. The exploratory research question was as follows: Is there an association between achievement motivation, OCB, and perceived stress at baseline and leadership style at week 7 and week 11?

## 2. Methods

### 2.1. Procedure and Study Design

The procedure and the study design have been extensively described elsewhere [58]. Briefly, we performed the study during autumn 2011 and spring 2012; we approached recruits attending the Swiss Armed Forces Infantry School of Aarau (Switzerland) during their BMT (duration: 21 weeks) and asked them to participate in the present online-run study on the longitudinal relation between leadership style (transformational; transactional; laissez-faire) and psychological variables such as achievement motivation, OCB, and perceived stress. To this end, we informed participants about the aims of the study and

the confidential data handling. Recruits were thoroughly informed that participation or non-participation would have neither a favorable nor unfavorable influence on their BMT qualifications.

At the start of their BMT (baseline), participants completed a series of online questionnaires covering sociodemographic information, achievement motivation, OCB, and perceived stress (see details below). At week 7, participants rated the leadership style of their leaders; at week 11, participants once again completed questionnaires on achievement motivation, OCB, and perceived stress, and they rated the leadership style of their leaders (see details below). At week 11 of the Basic Military Training, recruits were assigned to those who attended the NCO (non-commissioned officer) school, and those who completed the Basic Military Training as soldiers. As such, simply for logistic and organizational reasons, the decision was to accomplish the assessments as long as the entire group of recruits was available.

The Ethics Committee of the Canton of Aargau (Aarau, Switzerland; AGEK (Arbeitsgemeinschaft der Schweizerischen Forschungs-Ethikkommission für klinische Versuche, Aargau, protocol code 2011/008) approved the study, which was performed in accordance with the seventh and current [59] edition of the Declaration of Helsinki.

The present analysis was part of a larger study of biological and psychological characteristics of recruits to the Swiss Armed Forces [50,51,56,60–62]. Data have not previously been published or presented in this form.

### 2.2. Participants

As described previously [58], the inclusion criteria for the present study were: male sex; compliance with the study conditions, specifically, being willing and able to complete questionnaires in German; signed written informed consent. Exclusion criteria were: clicking through the items either only on the right or left side within a time lapse of some minutes, resulting in a standard deviation of zero on questionnaires with reverse-scored items (called "click-throughs").

A total of $n = 694$ recruits initially agreed to participate. Of these, 43 (6.20%) did not sign the written consent, 81 (11.67%) had French or Italian as their mother tongue, and 45 (6.48%) were identified as "click-throughs".

In conclusion, the final sample consisted of 525 participants (see Figure 1).

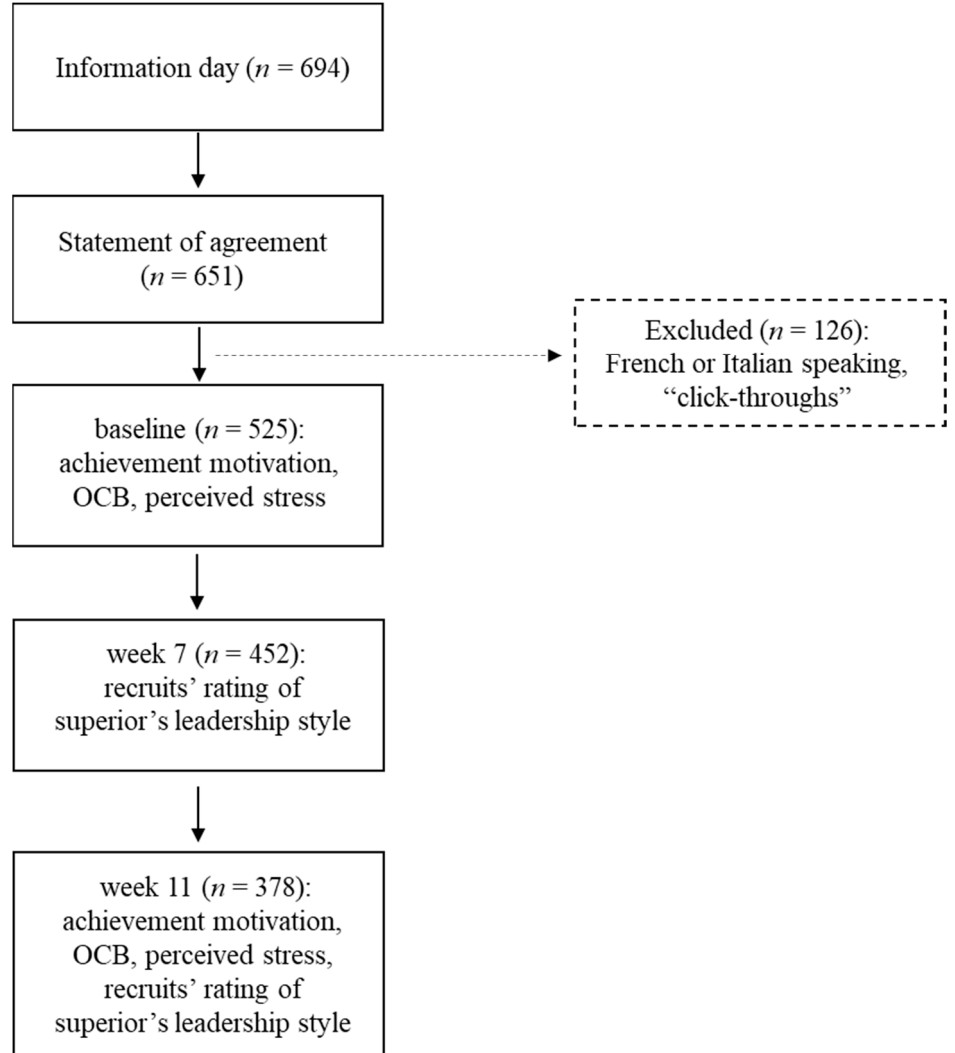

**Figure 1.** The flow chart shows the sample size of participants. At two stages (baseline and at week 11), self-rating data covering demographic information and psychological variables (i.e., achievement motivation, OCB, and perceived stress) were collected. At weeks 7 and 11, recruits additionally rated their superior's leadership style.

### 3. Measures

*3.1. Demographic Information*

Participants reported their age (in years), highest educational level (lower secondary school; upper secondary school; academic high school), mother tongue (German, French, Italian), and BMI.

*3.2. Leadership Style*

The leadership style was assessed using the multifactor leadership questionnaire (MLQ Form 5x Short) developed by Bass and Avolio [63,64], as translated into German by Felfe and Goihl [65] and validated on German samples by [66]. The German translation measures the nine dimensions of leadership (each dimension has 4 items; total of 36 items) grouped into three superordinate factors: transformational (idealized influence attributed, idealized influence behavior, inspirational motivation, intellectual stimulation, individual consideration), transactional (contingent reward, management by exception active, management by exception passive) and laissez-faire (avoidance/refusal of leading) leadership style. Recruits rated their leaders. Typical items are "is absent when needed", "provides me with assistance in exchange for my efforts", "spends time teaching and coaching", "avoids

making decisions". MLQ items are answered on a 5-point Likert scale from 1 (not at all) to 5 (frequently, if not always). Ratings were averaged as a mean value for each leadership style (Cronbach's $\alpha$: week 7: $\alpha = 0.917$, week 11: $\alpha = 0.946$).

### 3.3. Achievement Motivation

To assess achievement motivation, participants completed the achievement motivation inventory (German: Leistungsmotivationsinventar, LMI: [67]) using the 30 items short form LMI-K for evaluating a reliable Global Score, which were selected from the original 170 items and 17 facets [67]. Typical items are "when faced with a new job or task, I am often afraid of doing something wrong", "when I am determined to do something, and I don't succeed, then I do everything I can to still accomplish it" and "so that I will not be subject to criticism, I prefer to double my effort". Participants answered on a 7-point Likert scale ranging from 1 (strongly disagree) to 7 (strongly agree), with higher scores reflecting a more pronounced achievement motivation. The items were retained in the original version while the introduction was framed in military-specific terms (Cronbach's $\alpha$: Baseline $\alpha = 0.944$; week 11: $\alpha = 0.964$).

### 3.4. Organizational Citizenship Behavior

To assess recruits' OCB, participants completed the German translation [68] of Podsakoff, et al.'s [69] organizational citizenship behavior questionnaire. The scale includes five dimensions: conscientiousness, sportsmanship, civic virtue, courtesy, and altruism. Typical items are "I help others who have a heavy work load", "I am mindful of how my behavior affects other people's jobs" and "I am always ready to give a helping hand to those around me". Participants answered 25 items on 7-point Likert scales from 1 (strongly disagree) to 7 (strongly agree), with higher scores reflecting a more pronounced organizational citizenship behavior (Cronbach's $\alpha$: Baseline: $\alpha = 0.806$; week11: $\alpha = 0.844$).

### 3.5. Perceived Stress

Subjectively perceived stress was assessed with the German translation [70] of the perceived stress questionnaire (PSQ, [71]) over a time-period of four weeks and in four dimensions (joy, tension, demands, and worries). "You have many worries", "You feel tense", and "You feel safe and protected" are three typical items out of the 20 items of PSQ. Besides reverse scaling, items are answered on 4-point Likert scales ranging from 1 (=very seldom) to 4 (=almost ever). Higher scores reflecting higher perceived stress. The PSQ Indexes are mean values (Cronbach's alphas: Baseline: $\alpha = 0.74$; week 11: $\alpha = 0.76$).

### 3.6. Statistical Analysis

No sample size calculations were performed, as the basis of the present study is naturalistic and exploratory. Age, educational level, and BMI were examined as possible confounders; none of these variables systematically biased the scores for achievement motivation, OCB, perceived stress, or leadership style. Therefore, age, educational level, and BMI were not introduced as possible cofounders.

### 3.7. Kolmogorov–Smirnov Tests Were Performed to Show Normal Distribution of All Outcome Variables

To calculate the associations between achievement motivation, OCB, and perceived stress both at baseline and at week 11, and leadership style at week 7 and week 11, a series of Pearson's correlations was performed.

A series of *t*-tests were performed to compare means of achievement motivation and OCB between baseline and week 11.

A series of multivariate ANOVAs were computed with the factor Change in transformational leadership, that is: 1. Increase in superiors' transformational leadership from week 7 to week 11 of BMT; 2. No change or decrease in superiors' transformational leader-

ship from week 7 to week 11; dependent variables were: achievement motivation, OCB, and perceived stress at week 11.

A hierarchical linear regression was performed to predict the augmentation effect of transformational leadership over transactional leadership on study variables (achievement motivation, OCB, and perceived stress at week 11), controlling for baseline values. The residuals of the regression models were tested for normality.

The level of significance was set at $p < 05$. In some cases, the participants were unable to complete all of the questionnaires due to time constraints. Therefore, the sample size varies depending on time points, and are unequal in the different analyses, respectively.

All statistical procedures were performed with SPSS® 25.0 (IBM Corporation, Armonk, NY, USA).

## 4. Results

### 4.1. Sample Characteristics

All participants were male ($n$ = 525; mean age: 20.3 years ($SD$ = 1.16); BMI (mean BMI: 23.51 ($SD$ = 3.06)); 163 (31%) had completed lower secondary school, 203 (38.7%) had completed upper secondary school, and 159 (30.2%) had completed academic high school.

### 4.2. Achievement Motivation, Organizational Citizenship Behavior (OCB), and Perceived Stress at Baseline and at Week 11, and Leadership Style at Week 7 and Week 11

Table 1 provides the descriptive statistics and the correlations between achievement motivation, OCB, and perceived stress at baseline and at week 11 as well as leadership style at week 7 and week 11 as rated by recruits.

Higher scores for achievement motivation at baseline were associated with higher score for achievement motivation at week 11, and with higher OCB and lower perceived stress both at baseline and at week 11. Additionally, higher scores for achievement motivation at baseline were associated with higher scores for perceived transformational leadership and transactional leadership of superiors at week 7 and week 11.

Higher scores of OCB at baseline were associated with higher score of OCB at week 11, and with lower perceived stress both at baseline and at week 11. Additionally, higher scores of OCB at baseline were associated with higher scores for perceived transformational leadership and transactional leadership of superiors at week 7 and week 11.

Higher perceived stress at baseline was associated with higher perceived stress at week 11 and lower achievement motivation and OCB at week 11. Furthermore, higher perceived stress at baseline was associated with lower transformational leadership but not with transactional leadership both at week 7 and at week 11.

At week 7, higher transformational leadership was associated with higher transactional leadership both at week 7 and week 11, higher achievement motivation, OCB, and transformational leadership at week 11 and lower perceived stress at week 11. Furthermore, higher transactional leadership was associated with higher achievement motivation, OCB, transformational leadership, and transactional leadership at week 11, but not with perceived stress at week 11.

At week 11, the following results were found. Higher achievement motivation was associated with higher OCB and lower perceived stress, while achievement motivation was associated with higher transformational and transactional leadership. Higher OCB was associated with lower perceived stress, while OCB was associated with higher transformational and transactional leadership. Higher perceived stress was associated with both lower transformational leadership and lower transactional leadership.

Overall, higher transformational leadership at week 11 was associated with higher achievement motivation and OCB at baseline and at week 11 and lower perceived stress at baseline and at week 11. Higher transformational leadership at week 11 was associated with higher transactional leadership style at week 11.

**Table 1.** Descriptive statistics and correlations between achievement motivation, OCB, and perceived stress at baseline and at week 11 as well as leadership style at week 7 and week 11 as rated by recruits.

| | Baseline | | | Week 7 | | Week 11 | | | | | M | SD |
|---|---|---|---|---|---|---|---|---|---|---|---|---|
| | Ach. Motivation | OCB | Perceived Stress | Transfor-Mational Leadership | Trans-Actional Leadership | Ach. Motivation | OCB | Perceived Stress | Transfor-Mational Leadership | Trans-Actional Leadership | | |
| Sample size (*n*) | 524 | 452 | 520 | 452 | 452 | 372 | 377 | 361 | 378 | 378 | | |
| Baseline | | | | | | | | | | | | |
| Achievement motivation | - | 0.40 ** | −0.42 ** | 0.16 ** | 0.11 ** | 0.53 ** | 0.36 ** | −0.30 ** | 0.14 ** | 0.14 ** | 4.70 | 0.93 |
| OCB | | - | −0.33 ** | 0.50 ** | 0.23 ** | 0.54 ** | 0.71 ** | −0.45 ** | 0.38 ** | 0.24 ** | 4.93 | 0.66 |
| Perceived stress | | | — | −0.19 ** | −0.09 | −0.30 ** | −0.28** | 0.41 ** | −0.21 ** | −0.07 | 32.2 | 16.5 |
| Week 7 | | | | | | | | | | | | |
| Transform. leadership | | | | - | 0.49 ** | 0.23 ** | 0.38 ** | −0.26 ** | 0.50 ** | 0.29 ** | 3.60 | 0.61 |
| Transact. leadership | | | | | - | 0.11 * | 0.14 ** | −0.11 | 0.17 ** | 0.34 ** | 3.21 | 0.37 |
| Week 11 | | | | | | | | | | | | |
| Achievement motivation | | | | | | - | 0.66 ** | −0.41 ** | 0.37 ** | 0.38 ** | 4.74 | 0.94 |
| OCB | | | | | | | - | −0.46 ** | 0.43 ** | 0.30 ** | 4.88 | 0.71 |
| Perceived stress | | | | | | | | - | −0.32 ** | −0.16 ** | 41.5 | 14.4 |
| Transform. leadership | | | | | | | | | - | 0.59 ** | 3.38 | 0.76 |
| Transact. leadership | | | | | | | | | | - | 3.15 | 0.45 |

Notes. OCB = organizational citizenship behavior; * $p < 0.05$, ** $p < 0.01$.

*4.3. Correlations of the Study Variables Achievement Motivation and OCB between Baseline and Week 11*

Table 2 provides the descriptive and inferential statistical overview of achievement motivation and OCB and at baseline and week 11.

**Table 2.** Descriptive and inferential statistical overview of achievement motivation and OCB and at baseline and week 11.

|  | Time | | Statistics | |
|---|---|---|---|---|
| **Dimensions** | **Baseline** | **Week 11** | *t*-**Tests** | **Effect Sizes** |
| Sample size (*n*) | 359 | 359 | | Cohen's d |
|  | M (*SD*) | M (*SD*) | | |
| Achievement motivation | 4.72 (0.90) | 4.74 (0.94) | *t*(358) = −0.38 | 0.01 [T] |
| Sample size (*n*) | 364 | 364 | | |
|  | M (*SD*) | M (*SD*) | *t*(363) = 2.69 ** | 0.05 [T] |
| OCB | 4.95 (0.67) | 4.88 (0.71) | | |

Notes. ** $p < 0.01$; T = Trivial Effect Size; L = Large Effect Size.

As shown in Table 2, no significant differences of achievement motivation were observed between data of baseline and week 11. Furthermore, significant differences were shown of OCB (trivial effect size) between data of baseline and week 11.

*4.4. Achievement Motivation, Organizational Citizenship Behavior (OCB), and Perceived Stress at Week 11 among the Group Experiencing an Increase in Transformational Leadership between Week 7 and Week 11 and the Group Experiencing No Change or a Decrease in Transformational Leadership between Week 7 and Week 11*

Table 3 provides the descriptive and inferential statistical overview of variables achievement motivation, OCB, and perceived stress for recruits perceiving an increase in superiors' transformational leadership and for those perceiving no change or a decline in superiors' transformational leadership.

**Table 3.** Multivariate ANOVA for group comparisons of the group of recruits experiencing an increase of superiors' transformational leadership and the group of recruits experiencing constant or decreased superiors' transformational leadership from week 7 to week 11 of variables achievement motivation, OCB, and perceived stress at week 11.

|  | Group | | |
|---|---|---|---|
|  | **Increased** | **No Change or Decreased** | **Group** |
|  | Transformational leadership | Transformational leadership | |
| Degrees of freedom | | | 1 |
| *n* | 124 | 215 | |
|  | M (*SD*) | M (*SD*) | *F* partial eta$^2$ |
| Achievement motivation (week 11) | 4.87 (0.94) | 4.62 (0.93) | 5.48 * 0.02 [S] |
| Organizational citizenship behavior (week 11) | 5.04 (0.76) | 4.77 (0.66) | 11.74 ** 0.03 [S] |
| Perceived stress (week 11) | 39.35 (14.29) | 42.36 (14.62) | 3.37 0.01 [S] |

Notes. [S] = small effect size; * $p < 0.05$, ** $p < 0.01$; *M:* mean; *SD:* standard deviation; *F:* F-ratio.

All effect sizes were small. Descriptively, the group perceiving increased transformational leadership had the highest scores for achievement motivation and OCB and lowest scores for perceived stress (always at week 11).

### 4.5. Augmentation Effect of Transformational Leadership over Transactional Leadership on Its Influence on Achievement Motivation, Organizational Citizenship Behavior (OCB), and Perceived Stress at Week 11

To calculate the augmentation effect of transformational leadership over transactional leadership on achievement motivation, OCB, and perceived stress at week 11, a series of hierarchical regression analyses was performed.

### 4.6. Augmentation Effect on Achievement Motivation at Week 11

Table 4 provides the statistical overview of the long-term influence of transformational leadership style on achievement motivation at week 11, controlling for achievement motivation at baseline and transactional leadership style at week 11.

**Table 4.** Augmentation effect: The influence of transformational leadership style on achievement motivation (week 11) in a hierarchical regression analysis, controlling for achievement motivation (baseline) and transactional leadership style.

| | Achievement Motivation (Week 11) | | | | |
|---|---|---|---|---|---|
| **Variables** | **B** | **SE B** | **β** | **$R^2$** | **$\Delta R^2$** |
| Step 1 | | | | | |
| Ach. motivation (baseline) | 0.51 | 0.04 | 0.49 *** | 0.38 | 0.37 |
| Transactional leadership (week 11) | 0.66 | 0.09 | 0.32 *** | | |
| Step 2 | | | | | |
| Ach. motivation (baseline) | 0.49 | 0.04 | 0.47 *** | 0.40 | 0.39 |
| Transactional leadership (week 11) | 0.43 | 0.11 | 0.21 *** | | |
| Transformational leadership (week 11) | 0.23 | 0.06 | 0.19 *** | | |

Notes. $n$ = 358. *** $p < 0.001$; *B*: unstandardized beta; *SE B*: standard error for the unstandardized beta; *β*: standardized beta; $R^2$: R-squared; $\Delta R^2$: Delta-R-squared.

Higher achievement motivation at baseline and transactional leadership (week 11) predicted higher achievement motivation at week 11. Adding transformational leadership (week 11) to the model (step 2) increased the strength of the model to a modest but significant degree.

### 4.7. Augmentation Effect on Organizational Citizenship Behavior (OCB) at Week 11

A hierarchical regression analysis was conducted to test the long-term augmentation effect of transformational leadership style at week 11 on OCB at week 11, controlling for OCB at baseline and transactional leadership style at week 11. Table 5 provides the statistical overview.

**Table 5.** Augmentation effect: The influence of transformational leadership style on OCB (week 11) in a hierarchical regression analysis, controlling for OCB (baseline) and transactional leadership style.

| | OCB (Week 11) | | | | |
|---|---|---|---|---|---|
| **Variables** | **B** | **SE B** | **β** | **$R^2$** | **$\Delta R^2$** |
| Step 1 | | | | | |
| OCB (baseline) | 0.72 | 0.04 | 0.68 *** | 0.53 | 0.52 |
| Transactional leadership (week 11) | 0.22 | 0.06 | 0.14 *** | | |
| Step 2 | | | | | |
| OCB (baseline) | 0.68 | 0.04 | 0.64 *** | 0.54 | 0.54 |
| Transactional leadership (week 11) | 0.08 | 0.07 | 0.05 | | |
| Transformational leadership (week 11) | 0.15 | 0.04 | 0.17 *** | | |

Notes. $n$ = 360. *** $p < 0.001$; *B*: unstandardized beta; *SE B*: standard error for the unstandardized beta; *β*: standardized beta; $R^2$: R-squared; $\Delta R^2$: Delta-R-squared.

Higher OCB at baseline and transactional leadership at week 11 predicted higher OCB at week 11. Adding transformational leadership (week 11) to the model (step 2) increased the strength of the model to a modest but significant degree. However, the effect of transactional leadership on OCB was no longer significant.

### 4.8. Augmentation Effect on Perceived Stress at Week 11

Table 6 provides the statistical overview of the long-term influence of transformational leadership style on perceived stress at baseline, controlling for perceived stress at baseline and transactional leadership style.

**Table 6.** Augmentation effect: The influence of transformational leadership style on achievement motivation (week 11) in a hierarchical regression analysis, controlling for perceived stress (baseline) and transactional leadership style.

| Variables | Perceived Stress (Week 11) | | | | |
|---|---|---|---|---|---|
| | *B* | *SE B* | *β* | $R^2$ | $\Delta R^2$ |
| Step 1 | | | | | |
| Perceived stress (baseline) | 0.36 | 0.04 | 0.40 *** | 0.19 | 0.18 |
| Transactional leadership (week 11) | −4.33 | 1.57 | −0.14 ** | | |
| Step 2 | | | | | |
| Perceived stress (baseline) | 0.32 | 0.05 | 0.36 *** | 0.22 | 0.22 |
| Transactional leadership (week 11) | 0.22 | 1.92 | 0.01 | | |
| Transformational leadership (week 11) | −4.65 | 1.17 | −0.24 *** | | |

Notes. $n$ = 341. ** $p < 0.01$, *** $p < 0.001$; B: unstandardized beta; *SE B*: standard error for the unstandardized beta; *β*: standard-ized beta; $R^2$: R-squared; $\Delta R^2$: Delta-R-squared.

Higher perceived stress at baseline and lower transactional leadership (week 11) predicted higher perceived stress at week 11. Adding transformational leadership (week 11) to the model (step 2) increased the strength of the model to a modest but significant degree. On the other hand, the effect of transactional leadership on perceived stress was no longer significant.

### 4.9. Correlations between Achievement Motivation, OCB, and Perceived Stress at Baseline and Leadership Style at Week 7 and Week 11

As shown in Table 1, higher scores for achievement motivation and OCB at baseline were associated with higher scores for transformational and transactional leadership style both at week 7 and week 11. Additionally, higher scores for perceived stress at baseline were associated with lower scores for transformational leadership style both at week 7 and week 11, but not with transactional leadership.

## 5. Discussion

The key findings of the present study of Swiss military recruits during basic military training (BMT) are as follows. Achievement motivation and OCB scores did not noticeably differ from baseline to week 11. In addition, those recruits perceiving an increase in superiors' transformational leadership from week 7 to week 11 had higher scores for achievement motivation and OCB and lower perceived stress at week 11, though the effect sizes were small. Furthermore, transformational leadership augmented the effect of transactional leadership in predicting higher achievement motivation and OCB scores and lower perceived stress scores at week 11 (always controlling for baseline values). Through investigation, a reciprocal association was found between higher achievement motivation and OCB at baseline and higher transformational leadership and transactional leadership at week 7 and week 11. However, for perceived stress at baseline, the association was only found for transformational leadership at week 7 and week 11, but not for transactional leadership at both time points.

In our opinion, the present results expand upon the current literature in five ways.

1. First, the data show that achievement motivation and OCB appear constant over the first 11 weeks of BMT.
2. Second, transformational leadership is a supportive factor for achievement motivation, OCB, and perceived stress longitudinally.
3. Third, transformational leadership augments transactional leadership by predicting achievement motivation, OCB, and perceived stress longitudinally.
4. Fourth, the data show reciprocal association between achievement motivation, OCB, and perceived stress at baseline and transformational leadership at week 7 and week 11.
5. Fifth, these results were achieved for the first time during the basic military training of Swiss Army recruits who are legally obliged to perform this military service.

Three hypotheses and one explorative research question were formulated, and each of these is considered in turn.

With the first hypothesis, we expected that achievement motivation and OCB would not change between baseline and week 11 of BMT, and the results confirmed and expand upon earlier findings [35]. The present findings include OCB, as previous findings from Annen, Goldammer and Szvircsev Tresch [35], and supplement the achievement motivation. However, the present results extend upon the previous work insofar as these patterns were observed in Swiss military recruits who are fulfilling their civic duty in order to complete basic military training. It is likely that motivation in these circumstances would be lower than, for example, among males who had chosen their own employment. In addition, it turned out that motivational factors do not increase by themselves during BMT. This further accentuates the importance of leadership style.

With the second hypothesis, we assumed an increase in the perceived transformational leadership style of superiors from week 7 to week 11 would lead to higher achievement motivation and higher organizational citizenship behavior (OCB) scores and lower perceived stress in subordinates by week 11 of BMT. Again, this hypothesis was supported. We were thus able to replicate earlier findings [46,49,52–57]. However, the present research extends previous work in two ways. First, these links have not previously been observed for Swiss males undertaking their BMT. Our assumption is that in this case motivation was not so high as to have biased the pattern of results. Second, we note that most of the earlier studies in a military context have employed purely cross-sectional designs (e.g., [17]). In contrast, the present design was longitudinal.

In the third hypothesis, we expected that transformational leadership style would have an augmentation effect over transactional leadership longitudinally (at week 11) with respect to impact on achievement motivation, organizational citizenship behavior (OCB), and perceived stress in subordinates. This, in turn, was supported, making the present results consistent with previous results [4,17,47]. However, our study goes further in the following respects. First, we note that the earlier studies in a military context have employed purely cross-sectional designs [17], in contrast to our longitudinal design. Second, the present results demonstrate an augmentation effect on perceived stress in a military context.

Fourth, in the explorative research question we assumed, that the association between achievement motivation, OCB, and perceived stress at baseline and leadership style at week 7 and week 11 is reciprocal. The present data supports this assumption. Given this, it seems conceivable that recruits' specific personality traits could influence the perception of the leadership style of their supervisor during BMT. Others [72–74] stated that subordinates' psychological dimensions predicted the perception of leadership style. Particularly, higher achievement motivation and OCB and lower perceived stress lead to perceiving the superiors' leadership style as more transformational, as in most cases the relationship appeared to be stronger for transformational leadership than for transactional leadership. We note that the present results add to the current literature in that data were carried out from a longitudinal study design and in the military context. Future studies of military research should investigate the influence of recruit variables on the perception of leadership style.

Despite the replication and novelty of the present results, the following limitations warrant against overgeneralization. First, we assessed exclusively male recruits, though in contrast to other armies (e.g., U.S. Army), female recruits in the Swiss army are very few. Second, we assessed only a small number of male recruits carrying out their military duty; accordingly, sample biases cannot be ruled out. Given this, the generalizability of the present results remains unclear. Third, all data were collected by using self-reports. Fourth, due to the small to medium correlation coefficients, the variance in achievement motivation, OCB, and perceived stress remains largely unexplained, and the quality of the data did not allow a deeper understanding of the underlying psychological mechanisms. However, it is conceivable that compulsory military service could influence the current results with regard to motivational aspects (willingness to serve). Fifth, other important topics such as bullying [75], belonging to an ethnic minority [76], or the risk of unemployment after military service might have impacted on the results. Sixth, with regard to the group comparison, the results of the ANOVA showed only moderate differences (partial eta squared as effect sizes); given this, the present results should be overestimated. Seventh, the available data covered the first 11 weeks of the BMT, though it would be instructive to know if the achieved levels of motivation, OCB, and stress remained stable after military discharge. Eighth, while in the current study, an explanatory model of the longitudinal effects of transformative leadership on performance variables among Swiss recruits was investigated, in a future study, the applicability of this model might be also examined in a civil work environment. Ninth, to set up the theoretical framework from the present study we followed the literature on military leadership. It is conceivable, that focusing on pedagogical issues and publications, the theoretical background might yield to other conclusions. Tenth, it is conceivable that latent and unassessed confounders might have biased two or more dimensions in the same or opposite directions.

In BMT, the leadership interventions on subordinates are very frequent and intensive compared to everyday life in civilian working environments. Recruits are incorporated into tight leadership structures and permanent leadership influences [77]. Future studies of BMT in the Swiss Armed Forces might address the trainability of transformational leadership skills [66]. Research has shown that leadership interventions can have a positive effect on work outcomes [78,79]. Furthermore, other topics such as bullying [75] and willingness to undertake military service should be content for future military studies.

## 6. Conclusions

In the present study of a sample of Swiss men completing their compulsory BMT, transformational leadership appeared to be a valuable psychological construct associated with higher achievement motivation and OCB and lower perceived stress both cross-sectionally (at baseline) and longitudinally at week 11 of Swiss BMT. Furthermore, transformational leadership augmented transactional leadership in predicting achievement motivation, OCB, and perceived stress at week 11.

Regarding the theoretical implications of the present study, the following should be noted: The results were achieved for the first time during the basic military training of Swiss Army recruits who are legally obliged to perform this military service. Achievement motivation and OCB appear constant over the first 11 weeks of BMT. Transformational leadership is a supportive factor for achievement motivation, OCB, and perceived stress longitudinally and augments transactional leadership by predicting achievement motivation, OCB, and perceived stress longitudinally. Furthermore, data shows reciprocal association between achievement motivation, OCB, and perceived stress at baseline and transformational leadership at week 7 and week 11. For practical implications, it makes sense to integrate transformational leadership even more explicitly into the selection and training of leaders in the Swiss Armed Forces and to evaluate the relevant effects. The influence of achievement motivation, OCB, and perceived stress from the recruits' side on the perception of the superiors' leadership style should be considered for future studies.

In order to achieve the generalizability of our results, future research should include female participants, and a study design in the area of economics and private companies. In addition, a randomized experimental study design with intervention and control groups would allow to investigate in more details the variance of outcome variables. In addition, objective data measurements could be considered for physical stress measurements and objective performance indicators; this is to say: Leadership measurements can be assessed as a combination of self-assessment questionnaires, external assessment questionnaires completed by experts, and objective behavioral observations. Additionally, for future research, a long-term design with follow-up time points would allow to investigate mediators or moderators of the relationship of transformational leadership and achievement motivation, OCB, and perceived stress.

We mention the following limitations: The assessment included exclusively male recruits, self-reports, and integrated the first 11 of 21 weeks of BMT. Partially, results showed only small to medium correlation coefficients, and in group-comparison only moderate differences. Furthermore, unassessed confounders might have biased two or more dimensions in the same or opposite directions and the examined model should be proven in a civil work environment. Additionally, focusing on pedagogical issues and publications, the theoretical background might yield to other conclusions.

The positive importance of the present study can thereby be shown as these results were achieved for the first time during the basic military training of Swiss Army recruits who are legally obliged to perform this military service. The results show that motivational factors do not increase on their own during the first half of the BMT. The motivation of the recruits can, however, be increased and perceived stress decreased by the influence of transformational leadership. This is a very relevant finding for the leadership training in the Swiss Armed Forces.

**Author Contributions:** Conceptualization, S.S., M.P., R.L.M., T.W., L.R., H.A. and S.B.; methodology, S.S., M.P., R.L.M., T.W., L.R., D.S.-B., H.A. and S.B.; software, S.S., M.P., R.L.M., H.A. and S.B.; validation, S.S., M.P., R.L.M., T.W., L.R., H.A., S.B.; formal analysis, S.S., R.L.M., H.A., S.B.; investigation, S.S., M.P., R.L.M., T.W., L.R., D.S.-B., H.A. and S.B.; resources, S.S., M.P., R.L.M., T.W., L.R., H.A. and S.B.; data curation, S.S., M.P., R.L.M., T.W., H.A. and S.B.; writing—original draft preparation, S.S., H.A. and S.B.; writing—review and editing, S.S., M.P., R.L.M., T.W., L.R., D.S.-B., H.A. and S.B.; supervision, S.S., R.L.M., H.A. and S.B.; project administration, S.S., M.P., R.L.M., T.W., L.R., H.A. and S.B.; funding acquisition, S.S., M.P., T.W., L.R. and H.A. All authors have read and agreed to the published version of the manuscript.

**Funding:** The Swiss Federal Department of Defense, Civil Protection and Sport (DDPS) and the Military Academy of the Swiss Federal Institute of Technology ETH Zurich, Switzerland, supported this study financially. The views expressed are solely those of the authors. They do not reflect the official policy or position of the Swiss Armed Forces, the Swiss Federal Department of Defense, or the Swiss Government.

**Institutional Review Board Statement:** The study was conducted according to the guidelines of the Declaration of Helsinki and approved by the Ethics Committee AGEK (Arbeitsgemeinschaft der Schweizerischen Forschungs-Ethikkommission für klinische Versuche, Aargau, protocol code 2011/008; date of approval: 8 April 2011).

**Informed Consent Statement:** Informed consent was obtained from all subjects involved in the study.

**Data Availability Statement:** Data belong to the Swiss Armed Forces and to the Swiss Federal Office of Sport; data are made available to experts in the field upon request and upon submission of a detailed description of the reason for the request.

**Acknowledgments:** The authors would like to acknowledge the entire study team. Special thanks go to the team and the recruits of the Armed Forces Infantry School of Aarau, Switzerland. Further, we thank Nick Emler (University of Surrey, Surrey, UK) for proofreading the manuscript.

**Conflicts of Interest:** The authors declare that there is no conflict of interest.

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
