# Peer review of "Transformational Leadership, Achievement Motivation, and Perceived Stress in Basic Military Training: A Longitudinal Study of Swiss Armed Forces"

_sustainability, doi:10.3390/su132413949_

Round 1

Reviewer 1 Report

This paper is interesting, has scientific sounding. But I would like to say some critical positions.

My first observation: in my point of view, the title of the article is inappropriate. It does not show the idea of the problem; this is just a statement.  
The main  requirement when we start to formulate the title of the paper: we need to provide an object + problem.

How did the main keywords of this paper "Superiors’ transformational leadership" have the sounding to the research design-logic? I do not find, sorry.

On other hand, the authors statement ( p. 4) , that "While the impact of transformational leadership on stress has been studied  in numerous civilian organizations, there has been no corresponding work in military organizations." is not correct_ the are some papers in Europeans countries addressing this object.

Another point: research question should be constructed.

The following statement, which the authors make in the article as a research question (p. 4), is incorrectly worded: this is not the research question, it is a statement of main finding: " Exploratorily, we investigated the association between achievement motivation, OCB and perceived stress at baseline and leadership style at week 7 and week 11." 

The theoretical part of this paper lacks the development of the concept itself: on Leadership Behaviour (concept of leadership, and recently there has been an increase in new leadership terminology and definitions in the academic literature: Collaborative leadership; Entrepreneurial leadership; Chameleon leadership; Complex leadership/ Some authors discussed on leadership behaviours, on which innovative behaviour of subordinates (in this case military officers) depends.

Conclusions???

WE CAN SEE A STUDY/RESEARCH REPORT, BUT NOT A SCIENTIFIC ARTICLE.

Author Response

We thank Reviewer #1 for their valuable comments, which helped us to improve the quality of the manuscript. Please find the detailed point-by-point-attached as a separate file. 

Again, thank you very much for all your kind efforts. 

Reviewer 2 Report

Review Report – Sustainability

The paper investigates the link between transformational leadership styles and a series of psychological variables in a Swiss military setting. Since the context is the mandatory 21-week basic military training course taken by all Swiss conscripts the setting is, in my opinion, more training than military, which implies that the paper is more concerned with pedagogical issues rather than military leadership. Please comment!

Abstract

The abstract could be significantly shortened.

Introduction

Please add a reference after the first assertion in the introduction section. (First sentence in the introduction)

Methods

Why did you not follow the recruits for the entire 21 weeks of the BMT? This design decision needs to be elaborated upon in the methods section.

Make the used questionnaires available, either as a supplementary file or as an addendum, either in German or in English. This makes replication easier, and it also makes the study more transparent.

Statistical Analysis. Please add relevant references for natural and exploratory studies, and for possible cofounders.

Author Response

We thank Reviewer #2 for their valuable comments, which helped us to improve the quality of the manuscript. Please find the detailed point-by-point-attached as a separate file. 

Again, thank you very much for all your kind efforts. 

Round 2

Reviewer 1 Report

The authors sought to change the article; in response to reviewers ’comments; this is great!

The title of the paper now is sufficient; just one modification can be done_

The positive impact of transformational leadership on recruits’ motivational aspects and perceived stress in Basic Military Training: a longitudinal study of the Swiss Armed Forces  

Abstract section now is good.

One critical point, sorry:)

The development of methodological trajectories of innovation management /leadership/ and leadership behaviour are not sufficient described.

Another critical observation regarding the construction of conclusions (the following structure is recommended): 

  • Main findings. 
  • Theoretical implications
  • Practical implications
  • The limitation of the study 
  • The positive importance ....

However, I respect the hard work and efforts of the authors and reserve the right to the editor for final evaluation.

Author Response

Again, we thank Reviewer #1 for the care devoted to thoroughly deal with the revision. 

As requested, and based on Reviewer #2' suggestions, the text was further modified and improved.

Please find the detailed point-by-point-response and the modified manuscript attached as separate files.

This manuscript is a resubmission of an earlier submission. The following is a list of the peer review reports and author responses from that submission.